# Clinical, Immunological, and Functional Characterization of Six Patients with Very High IgM Levels

**DOI:** 10.3390/jcm9030818

**Published:** 2020-03-17

**Authors:** Vera Gallo, Emilia Cirillo, Rosaria Prencipe, Alessio Lepore, Luigi Del Vecchio, Giulia Scalia, Vincenzo Martinelli, Gigliola Di Matteo, Carol Saunders, Anne Durandy, Viviana Moschese, Antonio Leonardi, Giuliana Giardino, Claudio Pignata

**Affiliations:** 1Department of Translational Medical Sciences-Section of Pediatrics, Federico II University, 80131 Naples, Italy; veragallo86@hotmail.com (V.G.); emiliacirillo83@gmail.com (E.C.); prencipe.rosaria88@gmail.com (R.P.); giuliana.giardino@unina.it (G.G.); 2Department of Molecular Medicine and Medical Biotechnology, Federico II University, 80131 Naples, Italy; alessiolep@gmail.com (A.L.); leonardi@unina.it (A.L.); 3Advanced Biotechnologies s.c.a.r.l., CEINGE, 80131 Naples, Italy; luigi.delvecchio@unina.it (L.D.V.); scalia@ceinge.unina.it (G.S.); 4Department of Clinical Medicine and Surgery-Hematology, Federico II University, 80131 Naples, Italy; vincenzo.martinelli@unina.it; 5Department of Systems Medicine, University of Rome Tor Vergata, 00133 Rome, Italy; di.matteo@med.uniroma2.it; 6Center for Pediatric Genomic Medicine, Children’s Mercy Hospital, Kansas City, MO 64108, USA; csaunders@cmh.edu; 7Laboratory of Human Lymphohematopoiesis, INSERM UMR U1163 and Paris Descartes—Sorbonne Paris Cité University, Imagine Institute, 75743 Paris, France; anne.durandy@inserm.fr; 8Pediatric Allergology and Immunology Unit, University of Rome Tor Vergata, Policlinico Tor Vergata, 00133 Rome, Italy; moschese@med.uniroma2.it

**Keywords:** hyper IgM syndrome, lymphoproliferative disorders, class switch recombination, somatic hypermutation, DNA repair

## Abstract

Very high IgM levels represent the hallmark of hyper IgM (HIGM) syndromes, a group of primary immunodeficiencies (PIDs) characterized by susceptibility to infections and malignancies. Other PIDs not fulfilling the diagnostic criteria for HIGM syndromes can also be characterized by high IgM levels and susceptibility to malignancies. The aim of this study is to characterize clinical phenotype, immune impairment, and pathogenic mechanism in six patients with very high IgM levels in whom classical HIGM syndromes were ruled out. The immunological analysis included extended B-cell immunophenotyping, evaluation of class switch recombination and somatic hypermutation, and next generation sequencing (NGS). Recurrent or severe infections and chronic lung changes at the diagnosis were reported in five out of six and two out of six patients, respectively. Five out of six patients showed signs of lymphoproliferation and four patients developed malignancies. Four patients showed impaired B-cell homeostasis. Class switch recombination was functional in vivo in all patients. NGS revealed, in one case, a pathogenic mutation in *PIK3R1*. In a second case, the *ITPKB* gene, implicated in B- and T-cell development, survival, and activity was identified as a potential candidate gene. Independent of the genetic basis, very high IgM levels represent a risk factor for the development of recurrent infections leading to chronic lung changes, lymphoproliferation, and high risk of malignancies.

## 1. Introduction

High IgM levels are found in rare immunological disorders, such as hyper IgM (HIGM) syndromes, but also in autoimmune or acquired infectious diseases [1,2,3]. Elevated monoclonal IgM levels are also found in B-cell lymphoproliferative disorders (LPD) [4,5]. HIGM syndromes are characterized by normal or increased IgM levels and absent or strongly reduced levels of the other immunoglobulin isotypes due to impaired class switch recombination (CSR), which is sometimes associated with abnormal somatic hypermutation (SHM). Both processes require the integrity of cell DNA repair machinery, although with different repair mechanisms. Within the genetic disorders, so far, six genes have been implicated, coding for molecules involved in CD40 and CD40 ligand signaling (CD40 and CD40L), or in cytosine and cytidine deaminase process (AID) [6,7,8]. Traditionally, HIGM syndromes are classified in Online Mendelian Inheritance of Man (OMIM) in five groups. HIGM type 1 (OMIM # 308230) is an X-linked (XL) disorder due to CD40L deficiency, accounting for about 70% of the forms with known genetic basis [9,10]. HIGM type 2 (OMIM # 605258) which is due to mutations in *AID* gene, type 3 (OMIM # 606843) which is caused by *CD40* defects, and type 5 (OMIM # 608106) which is caused by mutations in the uracil DNA glycosylase (*UNG*) are inherited in an autosomal recessive manner. HIGM type 4 (OMIM # 608184) is a condition with an undefined genetic cause, phenotypically similar to type 2 [3]. An autosomal-dominant form due to mutations in the C-terminal region of AID has been described in a few patients [11]. The prevalence of these conditions varies in different ethnicities. Globally, HIGM constitute 0.3% to 2.9% of all PIDs [12]. The estimated frequency of CD40L deficiency is 2:1,000,000 in males [13]. Autosomal recessive forms are very rare. AID deficiency is estimated to affect less than 1:1,000,000 individuals [14]. CD40 deficiency is more common than AID deficiency, however, the real incidence cannot be calculated since the majority of the patients have not been described [10,12]. Other HIGM syndromes are caused by mutations, in post-meiotic segregation 2 (PMS2) and INO80 genes [15,16,17]. CD40L deficiency represents the most common genetic cause of HIGM syndrome. Clinical phenotype, pathogenesis, and prognosis related to the treatment of this specific form are well characterized. Elevated IgM levels were also reported in other well-defined immunodeficiencies, such as nuclear factor κB (NF-κB) essential modulator (NEMO), recombination-activating gene (RAG) 2, lipopolysaccharide-responsive beige-like anchor (LRBA), ataxia telangiectasia mutated (ATM), ARTEMIS, and dedicator of cytokinesis (DOCK) 2 deficiency [3,18,19,20,21,22,23]. Most of these conditions are characterized by increased susceptibility to malignancies and infections. Recently, novel genetic defects responsible for primary immunodeficiencies (PIDs) with HIGM have been characterized, thanks to the introduction of next generation sequencing (NGS) technologies in clinical settings. However, the genetic origin remains undefined in about 30% of the cases [1]. Clinical manifestations are variable and peculiar clinical features which are related to specific genetic alterations. However, independent of the genetic alteration, the overall clinical phenotype of HIGM is usually characterized by increased susceptibility to recurrent bacterial and opportunistic infections associated with autoimmunity and cancer susceptibility. The aim of this study is to characterize clinical phenotype, immune impairment, and functional phenotype in six patients with very high IgM levels, in which classical HIGM syndromes were ruled out. In particular, we focus on differences and commonalities with XL-HIGM and other well defined PIDs presenting with high IgM levels that help to predict the prognosis and improve the management of patients with less characterized forms. Moreover, we also try to shed light on the pathogenic mechanisms leading to HIGM in these patients.

## 2. Experimental Section

### 2.1. Patients

Six patients (four female) from unrelated non-consanguineous families with high IgM levels (range 416–1060 mg/dL) were enrolled in the study, after signed informed consent. CD40L alterations were ruled out through direct Sanger sequencing or flow cytometry in all patients. Normal cytofluorimetric expression was considered sufficient to rule out a *CD40L* genetic defect, especially in female subjects. Direct Sanger sequencing also ruled out alterations of the following genes in selected cases, during the followup: *CD40* (Patient 5 (P5) and Patient 6 (P6)), *AICDA* (Patient 1 (P1) and Patient 4 (P4) to P6), *UNG* (P1, P5 and P6), *NEMO* (P1), *TNFRSF13B* (P1), and *ATM* (P1). The study was carried out in accordance with the Declaration of Helsinki and approved by the institutional Ethical committee “Carlo Romano” of Federico II University (project identification codes 25/16 and 162/16). 

### 2.2. B-Cell Immunophenotyping and CD40-CD40L Expression

Whole blood anticoagulated with EDTA was used for multicolor flow cytometry immunophenotyping and processed within 24 h. Cells were exposed to directly conjugated mouse anti-human monoclonal antibodies (mAb) using the following fluorochrome conjugated antibodies: anti-CD45 peridinin-chlorophyll (PerCP) or anti-CD45 horizonV500 (HV500), anti-IgD phycoerythrin (PE), anti-IgM allophycocyanin (APC), and anti-IgG phycoerythrin (PE) all from BD Biosciences (Becton Dickinson, San Jose, CA, USA); anti-CD19 allophycocyanin (APC) and anti-CD27 FITC all from Invitrogen (Invitrogen, Carlsbad, CA) and Caltag (Caltag Medsystems, Buckingham, UK). The cells were incubated with directly labelled antibody at 4 °C in the dark for 30 min, washed and resuspended in 4 mL of NH4Cl, and then washed in PBS. Flow cytometric analysis was performed on a BD FACS Canto II flow cytometer (Becton Dickinson, San Jose, CA, USA) and the analysis was performed using BD FACSDiva software. Lymphocytes were identified by gating on viable CD45+ cells or on FSC and SSC and B-lymphocytes were gated on CD19+ cells. After 48 h incubation with PMA and ionomycin, CD40L and CD40 staining was performed through directly conjugated mouse anti-human mAb.

### 2.3. Study of CSR and SHM In Vitro

Peripheral blood mononuclear cells (PBMCs) were cultured in the presence of 500 ng/mL of soluble CD40-L and 100 U/mL of rIL-4. Proliferation was assessed by measuring [3H] thymidine uptake. IgE production was evaluated in supernatants by ELISA on day 12 [24]. SHM generation in the VH3-23 region of IgM on purified CD19+CD27+ B cells was performed in two patients, as previously described [8]. VH3-23 region was chosen since V3-23 Ig V gene is expressed in 4% to 10% of B cells [25]. PBMCs were isolated by flow cytometry using FITC-anti-CD19 and PE-anti-CD27 mAb (Invitrogen, Carlsbad, CA, USA). RNA was purified with Trizol reagent and cDNA was obtained by reverse transcription with an oligo dT primer. PCR was carried out with Pfu polymerase (PfuTurbo, Stratagene) (Sigma-Aldrich, St. Louis, MO, USA) using primers for the VH3-23 leader exon (5′-GGCTGAGCTGGCTTTTTCTTGTGG-3′) and Cµ region (5′-TCACAGGAGACGAGGGGGAA-3′) (35 cycles at 94-C for 45 s, 60-C for 1.5 min, and 72-C for 2 min). After amplification by high fidelity Pfu, addition of 3′ A-overhangs post amplification was performed by adding Taq polymerase to the PCR reaction, and incubating the mix at 72 °C for 10 min before proceeding to TA-cloning, following manufacture’s recommendation (Invitrogen, Carlsbad, CA, USA). VH3-23 positive colonies were sequenced with dRhodamine dye terminator cycle sequencing kit (Applied Biosystems prism, Foster City, CA, USA) and analyzed with Applied Biosystems prism 310 genetic analyzer. Frequency of mutations, expressed as a percentage of mutated nucleotides and all nucleotides, was evaluated for each clone on 300 nucleotides of the VH3-23 region including CDR1 and CDR2 domains. Polyclonality was established based on the diversity of the CDR3 of B cells bearing IgM. 

### 2.4. Whole Exome and Sanger Sequencing

Whole exome sequencing (WES) was performed in 5 cases (P6 died before). Since mutations in *PIK3CD* or *PIK3R1* genes have been recently reported as responsible for novel PIDs characterized by elevated IgM, normal/low IgG levels, lymphopenia, respiratory infections, lymph node enlargement, and elevated risk of lymphomas [26,27,28,29,30], these genes were analyzed by direct sequencing. Variants in causative or potential candidate genes identified through WES were confirmed through Sanger sequencing. Genomic DNA was isolated from peripheral blood lymphocytes using QIAamp DNA Blood Mini Kit (Qiagen, Hilden, Germany). Samples were sequenced to at least 2.5 GB on Illumina MiSeq with TruSeq MiSeq V3 reagents, yielding paired 250 nucleotide reads. Samples for exome sequencing were prepared using TruSeq HT library preparation kit (Illumina; San Diego, CA, USA) and, subsequently, exome enrichment was performed through xGen Exome Research Panel V1.0 (Integrated DNA Technologies, Coralville, IA, USA) according to the manufacturer’s protocols. Bioinformatic analysis was carried out, as previously described [31,32].

### 2.5. Real-Time qRT-PCR of ITPKB Gene

RNA from Patient 3 (P3) and controls’ PBMCs was extracted using Trizol reagent (Invitrogen, Carlsbad, CA), and Phase-lock gel columns (Eppendorf, Amburgo, Germany) according to manufacturer’s instructions. RNA was reverse transcribed by Transcriptor First Strand cDNA Synthesis Kit (Roche Applied System, Mannheim, Germany). qPCR reactions were performed in duplicate. Amplification of cDNAs was performed using SYBR Green and analyzed with Light Cycler480 (Roche Applied Systems, Mannheim, Germany). Cycling conditions comprised an initial denaturation at 94 °C for 5 min, a phase of annealing/extension specific for each gene. A dissociation procedure was performed to generate a melting curve for confirmation of amplification specificity. Primer sequences for ITPKB were the following: Primer A forward, 5′-GCACTGGTCTCCCTTCGTC-3′; reverse, 5′-CCAGGTAGGTCCTGATTCCC-3′; Primer B forward, 5′-GGACACGCAGGGAGTTTCAAG-3′; reverse, 5′-TCGTCCATCTGGTTGTAGCG-3′. The primer sequences for Actin were the following: forward 5′-GATCAAGATCATTGCTCCTCCTG-3′ and reverse, 5′-AGGGTGTAAAACGCAGCTCA-3′. Results are expressed as mean + standard error (SE) of 3 repeated experiments and each gene expression was normalized to β-actin as the housekeeping gene. The relative transcript abundance was represented as -DCt = (Ct gene - Ct reference) and the relative changes in gene expression was analyzed using the 2-DDCt method. Data presented as the means ± SD were analyzed with a Student’s t test analysis. *P* values <0.01 were considered significant.

## 3. Results

### 3.1. Clinical and Immunological Data of HIGM-Like Patients

As shown in Table 1, five out of six patients studied suffered from recurrent bacterial and viral infections, in particular, of the upper and lower respiratory tract, causing in P1 a significant lung damage characterized by bronchiectasis and atelectasis. Atelectasis were also found in P4, whereas interstitial lung disease was documented in Patient 2 (P2). P1 and P2 also experienced complicated viral infections, including measles complicated by pneumonia (P1) and chickenpox with severe ocular involvement (P2). Quantitative PCR for EBV, CMV, HCV-RNA, and HBV-DNA, was negative in patients with hypogammaglobulinemia who underwent intravenus immunoglobulin replacement therapy (IVIG) (P1, P4–P6). Involvement of the reticulo-endothelial system with lymphadenopathy, liver and spleen enlargement was documented in five patients. Three patients had autoimmunity or signs of chronic inflammation, i.e., alopecia (P4), autoimmune myofascitis and recurrent fever (P1, P2). Other clinical features included urticarial-like lesions or other recurrent skin lesions in four cases, growth retardation and delayed puberty in two cases, mood and behavioral disorders in two cases, and Arnold–Chiari malformation in one patient. 

In four cases, IgG levels were 2 SD below the mean value of age-matched controls, requiring IGIV, as shown in Table 2; IgA isotypes were also below reference values in P1, P5, and P6. 

Four patients developed B-cell LPD during the followup. P1 was diagnosed, at age 11 years, with non-Hodgkin marginal zone B-cell lymphoma with a cutaneous infiltrate which was, subsequently, treated with hematopoietic stem cell transplantation (HSCT) from a matched related donor. P2 was diagnosed at the age of 47 years with a low-grade mucosa-associated lymphoid tissue (MALT) lymphoma associated with Helicobacter pylori infection, whose eradication induced MALT remission. P5 received successful chemotherapy for a cervical Hodgkin lymphoma at the age of 19 years. P6 was diagnosed with a diffuse large B-cell lymphoma, invading the colonic mucosa, at the age of 17 years but he died from infectious complications, as previously reported [33]. A long-term longitudinal evaluation of IgM levels well before the onset of LPD was available for P1 and P2, as illustrated in Figure 1. Both patients had elevated IgM levels, ranging from 680 and 750 mg/dL in P1 and from 910 to 1060 mg/dL in P2. In both patients, no significant further increase in IgM levels was observed when LPD was diagnosed, while a significant reduction was documented in P1 in the subsequent four years post chemotherapy (range 280–386 mg/dL). 

In both P5 and P6, elevated IgM levels were documented in the first years of life on the occasion of an immunological evaluation required for recurrent bacterial infections and persisted high at the time of LPD onset. An additional patient with a clinical phenotype characterized by elevated IgM, hypogammaglobulinemia, autoimmune manifestations, and LPD was identified but not included in the study since most of the immunological evaluations were not performed before the onset of LPD. In P5 and P6, at the time of LPD diagnosis, immunological evaluation also revealed lymphopenia, predominantly involving the CD4 population. In the remaining patients, no significant abnormalities were found in the T-cell compartment (Table 2).

### 3.2. B-Lymphocyte Profiling and CD40-CD40L Expression

Total CD19+ B lymphocytes percentage was reduced in P2, P3, P5, and P6 (5%, 2%, 4%, and 0%, respectively) [34]. CD19+CD20-IgG+ mature B cells were absent in all the studied patients. Circulating CD19+CD27-IgD+ B cells (naive B cells) were normal or low/normal in all the patients investigated (P1–P4), implying a normal central B-cell development. CD19+CD27+IgM+ memory B cells were significantly reduced in all the patients studied (range 1.6–5% of CD19+cells) [35]. As illustrated in Table 2, three of the four patients studied also showed a significant reduction of CD19+CD27+IgM- switched memory B cells (0%, 2.9% and 0% of CD19+cells). Intriguingly, as shown in Figure 2, P1 and P2, who developed LPD, showed the most significant alteration in switched memory B-cell population. However, these patients exhibited a normal capability to produce specific IgG antibodies. Unfortunately, B-lymphocyte profiling was not available for P5 and P6. 

CD40L and CD40 expression was normal in all cases except for P3, in whom CD40L expression was reduced. In particular P3 CD4+ T cells were not able to upregulate CD40L expression after stimulation, differently from what was observed in cells from a healthy control (0.2% vs. 2.7%) (Figure 3). CD40LG gene alterations were ruled out by direct sequencing in this patient. 

### 3.3. Reduced SHM with Normal In Vivo and In Vitro CSR

As shown in Table 3, IgG specific antibodies were detectable in four patients including two hypogammaglobulinemic patients before the start of replacement therapy (P1 and P4), indicating a normal in vivo CSR. It was not possible to study CSR in the remaining two cases, since they were already on IGIV at the time of the present study. CSR and SHM were studied in vitro in two patients (P1 and P2). After 12 days in culture with sCD40L+IL4, B cells were able to switch towards IgE in both the patients (Table 3). In addition, in these patients a normal B-cell proliferation following the exposure to IL4+sCD40L was also found (data not shown). 

The study of SHM, performed twice over a five-year period in P1, showed a slight reduction of the percentage of mutated CDR3-different clones in both patients as compared with healthy controls (9 out of 11, and 6 out of 10 in P1, and 6 out of 7 in P2 as compared with controls in whom nearly all clones were found mutated). Moreover, in both patients the frequency of mutations found in the VH3-23 region was lower than controls (P1, 1.9% and 1.2% and P2, 2.3% versus 3.5% to 6.3% of controls), while the nucleotide substitution pattern was normal (Figure 4).

### 3.4. NGS

WES allowed to achieve a definitive diagnosis only in P5, who showed a de novo, already reported, splice site mutation in PIK3R1 (c.1425+1G>T), also confirmed by direct sequencing [28,29]. Sanger sequencing of PIK3CD and PIK3R1 resulted normal in the four cases studied (P1, P2, P3, and P4). In P3, a heterozygous unreported frameshift variant in ITPKB gene (c.146_147insA), predicted to damage protein function with high confidence, was identified as a potential candidate gene (Figure 5). 

### 3.5. Real-Time qRT-PCR of ITPKB Gene from RNA 

Quantitative RT-PCR (qRT-PCR) from PBMCs showed that ITPKB gene expression was significantly lower in the P4 as compared with controls, as illustrated in Figure 6. Unfortunately, the patient did not authorize the execution of further studies aimed at the functional definition of the variation identified.

## 4. Discussion

We report on six patients with very high IgM levels in whom classical HIGM syndromes were excluded. Similar to other well-defined HIGM forms, the main clinical manifestations observed in these patients were recurrent respiratory infections, leading to the development of chronic lung changes and B-cell LPD. Most of the patients showed alterations of the B-cell maturation process, characterized by reduced memory and class-switched memory B cells, suggesting an impaired germinal center function. The defect of the CD19+CD27+IgM- switched memory subset was more pronounced in two of the four patients who developed LPD. B-cell abnormalities are usually observed in patients affected with CVID [36,37] and in other PIDs, including X-linked lymphoproliferative syndrome type 1 (XLP1), warts, hypogammaglobulinemia, infections, Myelokathexis (WHIM) syndrome, or DOCK8 deficiency [38,39,40]. Of note, CVID patients are also prone to develop lymphomas [41]. A significant correlation between high serum IgM levels at diagnosis and the development of either polyclonal lymphocytic infiltration or lymphoid malignancy has been reported in CVID [42]. In our cohort, most of the patients (five out of six) showed signs of lymphoproliferation and four of them developed B-cell LPD. This data confirms that, independent of the genetic basis, HIGM represents a risk factor for the development of LPD and patients with HIGM of any kind need a close follow-up for early identification of these conditions. 

CSR and SHM represent the two major maturation events required for an efficient humoral response, and both take place simultaneously in the germinal center after CD40 activation. Most of the genetically defined HIGM syndromes are due to alterations in genes involved in these processes. For this reason, we investigated the CSR and SHM of IgM on CD19+CD27+ B cells. Intriguingly, differently from classical HIGM syndromes, four patients showed protective antibody levels against different antigens, suggesting a normal in vivo CSR. In two patients, CSR was also studied in vitro and resulted normal. In the same patients, normal CSR was associated with a reduced frequency of SHM. It should be noted that the reliability of the test in these two patients could be affected by the low numbers of studied clones (due to the technique used and the decreased number of CD27+ cells), although it was confirmed twice in P1, supporting the hypothesis of a defective SHM process. It should also be noted that two patients developed chronic lung changes even in the presence of protective antibodies titers against different antigens, suggesting that the immunological memory can be a secondary lost in these patients or that the existing antibodies are not functioning properly. For this reason, IGIV should be timely started in these conditions even in the presence of residual protective antibody titers, in order to avoid the development of chronic lung changes. The impairment of SHM observed in these patients suggests alterations of germinal center functionality, as also suggested by the reduction of memory and class-switched memory B cells. Unfortunately, because of chemotherapy, HSCT, or death, we were not able to test more patients for SHM. However, we cannot exclude that alterations of the humoral immunity observed in these patients are due to a T-cell defect, even if four out of six patients had no obvious abnormalities of the T-cell phenotype. Decreased B-cell count, especially involving memory B cells, could also suggest a defect in B-cell proliferation or survival. The development of B-cell tumors in these patients suggests the presence of an intrinsic B-cell defect, likely associated with a DNA repair defect. DNA repair defects, as non-homologous end joining (NHEJ) defects or ataxia telangiectasia (A-T) [43] and Nijmegen breakage syndromes (NBS) [44], are characterized by defective in vitro CSR and normal frequency and pattern of SHM and are usually associated with a HIGM phenotype. Defects in base excision or mismatch repair mechanisms affect in vitro CSR, while SHM is usually normal in these conditions. In UNG- and MSH6-deficiencies [45,46] CSR alteration is associated with impaired SHM, with a dramatically abnormal pattern of nucleotide substitution. 

The genetic evaluation through WES achieved a definitive diagnosis in one case who showed an already described mutation in *PIK3R1*. Recently, heterozygous gain-of-function mutations in *PIK3R1*, encoding for p85α, one of the catalytic subunits of the PI3 kinase molecules, were reported as responsible for a novel form of immunodeficiency [28,29,30]. This novel immunodeficiency, similar to activated PI3 kinase delta syndrome (APDS) due to mutations of another subunit of the PI3K pathway, p110δ, is characterized by elevated IgM and low IgG serum levels, recurrent respiratory infections, lymph node enlargement, poor growth, and elevated risk to develop lymphomas. In a second patient, a variant of unknown significance (VUS) was identified in a potential novel candidate gene. The *ITPKB* gene variant has never been reported previously, even though a deletion involving this gene has been recently associated with a CVID phenotype with mood disorders [47]. Similar to the CVID patient described by Luis et al., our patient showed psychiatric symptoms and lymphopenia associated with recurrent otitis, without any infectious or eczematous skin lesions and recurrent sepsis. In our patient, quantitative RT-PCR documented a reduction of the RNA expression as compared with controls, suggesting that the variant identified affects the expression of the gene. Unfortunately, it was not possible to obtain a formal demonstration of the role of the VUS identified, since the patient did not consent to the execution of further investigations. The heterozygous status of the VUS made it difficult to model the variant in vitro. Recent evidence in murine models documents the role of this gene in T- and B-cell development, function, and survival [48,49]. Future studies on mouse models would help to clarify the pathogenic role of this VUS.

## 5. Conclusions

Taken together, our findings suggest that elevated polyclonal IgM levels, even in presence of a normal CSR recombination, is a warning sign for a B-cell disorder and should prompt clinicians to consider the risk of LPD in the investigation of patients affected. These patients can develop chronic lung changes even in the presence of residual protective antibody levels, suggesting that IGIV should be promptly started to avoid the development of complications. Moreover, an in-depth characterization of such patients at the molecular and functional level could lead to the identification of novel immunological pathways, paving the way to targeted therapy.

## Figures and Tables

**Figure 1 jcm-09-00818-f001:**
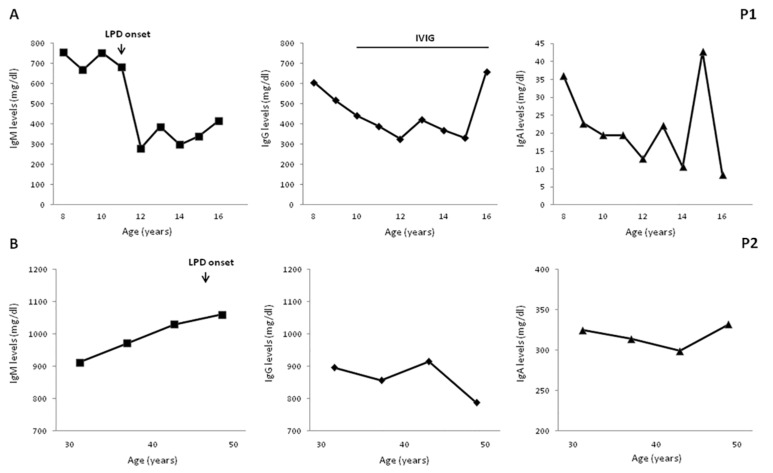
Long-term serum immunoglobulin levels evaluation. A long-term evaluation documented elevated IgM levels 5 and 10 years before lymphoproliferative disorder (LPD) in patient P1 (**A**); and in P2, respectively (**B**). IgG decreased in P1 post chemotherapy. Age-matched reference values in mg/dL (9–11 y) IgG 707–1919, IgA 60–270, and IgM 61–276; (12 to 16 y) IgG 604–1909, IgA 61–301, and IgM 59–297; (>18 y) IgG 737–1607, IgA 70–400, and IgM 40–230.

**Figure 2 jcm-09-00818-f002:**
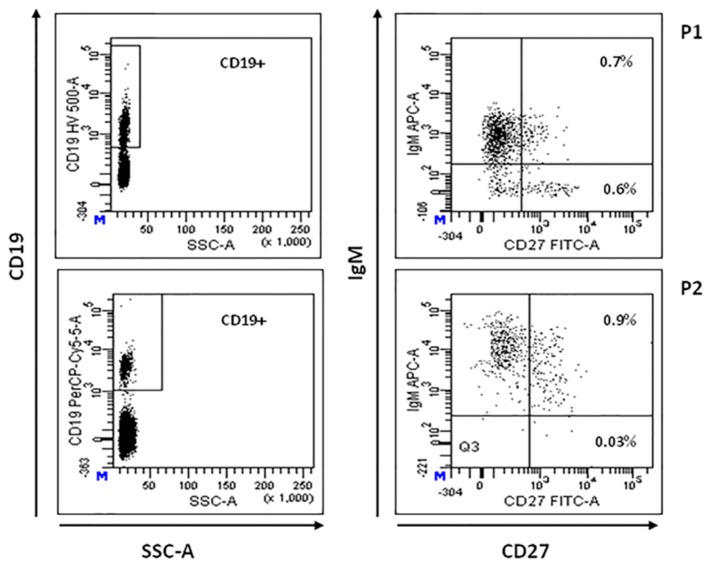
B-cell immunophenotyping. Representative flow cytometric plot showing total CD19+ cells (left panels), memory (CD19+ CD27+IgM+) and switched memory (CD19+ CD27+IgM-) B-cell populations in patients P1 and P2 (right panels), expressed as percentage of total lymphocyte.

**Figure 3 jcm-09-00818-f003:**
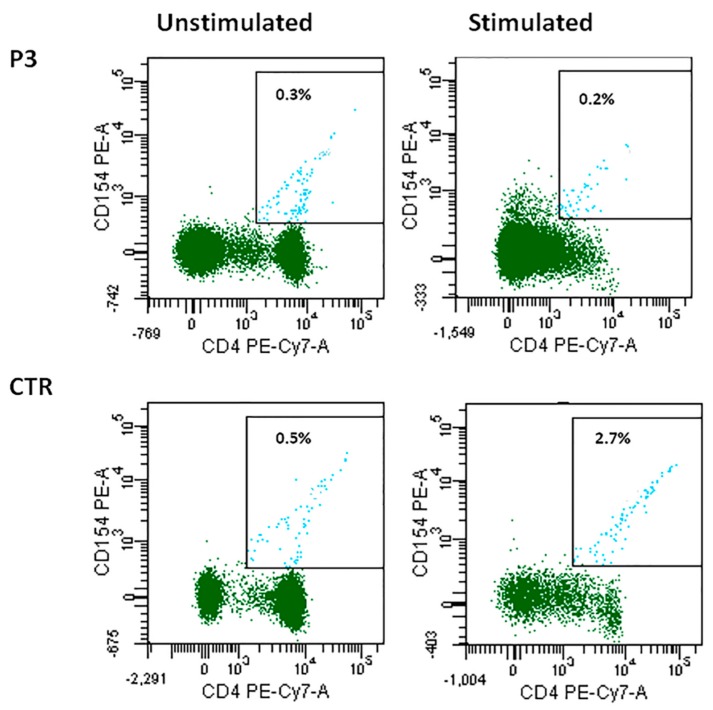
CD40L expression in patient P3 unstimulated and stimulated CD4+ cells as compared with the control.

**Figure 4 jcm-09-00818-f004:**
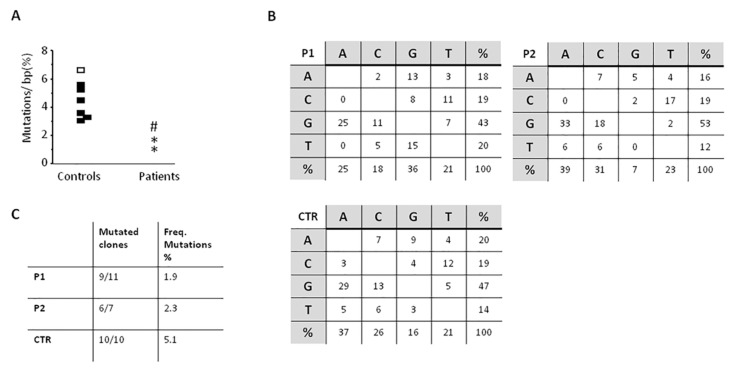
somatic hypermutation (SHM) results on the VH3-23 region of IgM on CD19+CD27+ isolated B cells. (**A**) Frequency of mutations on IgM+CD27+ B cells purified from controls and patients. Dots represent results for each subject. Controls: Black dots, all 10 clones mutated and white dots, 9/10 clones mutated. Patients: P1, * and P2, #; (**B**) Nucleotide substitution pattern for P1 (1st evaluation), P2, and control. The same pattern was observed in P1 in the 2nd evaluation. (**C**) The numbers of mutated clones from all studied clones is shown, as well as the frequency of mutations.

**Figure 5 jcm-09-00818-f005:**
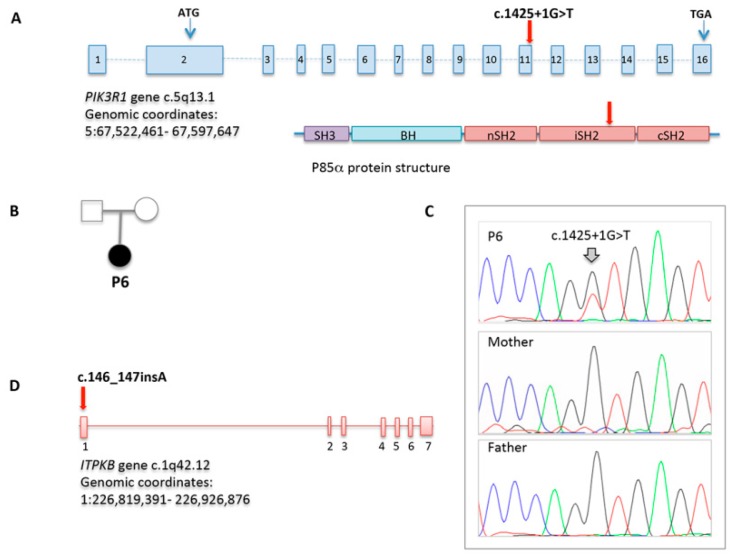
*PIK3R1* and *ITPKB* variations identified through WES. (**A**) *PIK3R1* gene structure which encodes p85α protein. The splice site mutation in patient P6 is indicated; (**B**) Pedigree of the family carrying the *PIK3R1* mutation; (**C**) Sequencing chromatograms in patient P6 and her parents; (**D**) *ITPKB* gene structure with the frameshift mutation identified in patient P4. Chromosome location and genomic coordinates are provided.

**Figure 6 jcm-09-00818-f006:**
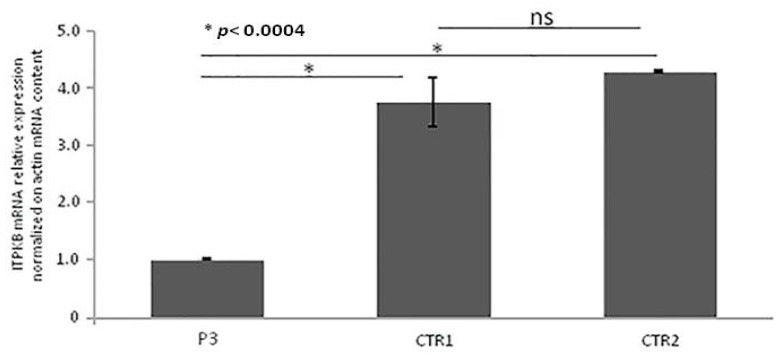
Real-time qRT-PCR of *ITPKB* gene mRNA expression. Quantitative RT-PCR was performed using two specific couples of primers. ITPKB mRNA expression was normalized on Actin mRNA content. The error bars represent technical replicates within a single experiment. Each experiment has been repeated three times. Statistical analysis was performed by comparison between controls and affected ITPKB mRNA content. *****
*p* < 0.0004 and ns = not statistically significant.

**Table 1 jcm-09-00818-t001:** Clinical features of the hyper IgM (HIGM)-like patients.

	P1	P2	P3	P4	P5	P6†
Age (y)	16	49	45	8	15	15
Sex	F	F	M	M	F	F
Clinical features						
Infections						
Bacterial	+	+	−	+	+	+
Opportunistic	−	−	−	−	−	−
Viral	+ *	+ **	−	−	−	−
Lung disease						
Bronchiectasis	+	−	−	−	−	−
Atelectasis	+	−	−	+	−	−
Interstitial lung disease	−	+	−	−	−	−
Lymphadenopathy	+	−	+/−	+	+	++
Autoimmunity	−	−	−	+	−	−
Inflammatory disease	−	+	−	−	−	−
Recurrent fever	−	+	−	−	−	−
Musculoskeletal involvement	+	+	−	−	−	−
Liver and/or spleen enlargement	+	−	+	+	−	−
Cutaneous manifestations	+	+	+	+	−	−
Cancer	NH- Lymphoma	MALT- Lymphoma	−	−	Hodgkin- Lymphoma	Diffuse large B-cell Lymphoma
Other			Mood disorder	Behavioral disorder	Growth and pubertal delay; bone defects; Arnold Chiari syndrome	Growth and pubertal delay

NH, non-Hodgkin; MALT, mucosa-associated lymphoid tissue; * P1 experienced a measles complicated by pneumonia; ** P2 experienced a chickenpox with severe ocular involvement; † patient died.

**Table 2 jcm-09-00818-t002:** Immunological and molecular findings of HIGM-like patients.

	P1		P2		P3	P4	P5	P6†
**Age (y)**	10 Before LPD	16 After LPD	34 Before LPD	49 After LPD	45	8	15 Before LPD	15 Before LPD
Immunological features								
IgG, mg/dL	442↓ *(IVIG)*	660	858	788	1500	294↓ *(IVIG)*	565↓ *(IVIG)*	140↓ *(IVIG)*
IgA, mg/dL	19↓	8.3↓	282	332	227	33	5↓	5↓
IgM, mg/dL	753↑	416↑	911↑	1060↑	516↑	800↑	443↑	596↑
Lymphocyte absolute counts/mL	7860	6950	*NK*	3.300	1400	3090	2100	2260
T-cell subsets								
CD3+ (% of lympho) (absolute value)	67 (5266)	72 (5004)	*NK*	79 (2607)	86 (1204)	74 (2286)	90 (1890)	95 (2147)
CD4+ (% of lympho)	23 (1807)	31 (2154)	*NK*	57 (1881)	48 (672)	40 (1236)	26↓ (546)	14↓ (300)
CD8+ (% of lympho)	40 (3144)	31 (2154)	*NK*	21 (693)	36 (504)	30 (927)	62↑ (1302)	70↑ (1502)
CD56+ (% of lympho)	2.7 (212)	4 (278)	*NK*	14 (462)	6 (84)	6 (185)	*NK*	*NK*
B-cell subsets								
CD19+ (% of lympho)	28.5↑ (2240↑)	25 (1737)	*NK*	5↓ (165↓)	2↓ (28↓)	12 (370)	4↓ (84↓)	0↓ (0)
CD19+CD27+IgM+ (IgM memory, % of CD19+)	12.3 (275)	3↓ (52.0↓)	*NK*	20 (33↓)	5↓ (1.4↓)	1.6↓ (5.9↓)	*NK*	*NK*
CD19+CD27+IgM- (switched memory, % of CD19+)	0↓ (0)	2.9↓ (50.3↓)	*NK*	0↓ (0)	10 (2.8↓)	12.5 (46.2)	*NK*	*NK*
Genetic alteration			*ITPKB* c.146_147insA		*PIK3R1* c.1425+1G>T			
Inheritance			*NK*		de novo			

LPD, lymphoproliferative disorders; IGIV, intravenous immunoglobulin; NK, not known. † patient died.

**Table 3 jcm-09-00818-t003:** In vivo and in vitro class switch recombination (CSR).

	P1	P2	P3	P4
In vivo CSR				
IgG anti-HbsAg	−	*ND*	−	−
IgG anti-Measles	+	+	*ND*	*ND*
IgG anti-CMV	+	+	+	+
IgG anti-EBV	+	+	+	+
IgG anti-VZV	+	+	+	+
IgG anti-Rubella	−	*ND*	+	*ND*
IgG anti-Mumps	−	*ND*		+
In vitro CSR				
IgE pg/mL (not stimulated)	173	2860	*ND*	*ND*
IgE pg/mL (stimulated)	14658	6120	*ND*	*ND*

ND, not done.

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
