# Peer review of "Clinical, Immunological, and Functional Characterization of Six Patients with Very High IgM Levels"

_jcm, 2020, doi:10.3390/jcm9030818_

Round 1
Reviewer 1 Report
The authors describe 6 patients who presented with very high IgM levels but whose clinical features did not seem to fit into the classical hyper IgM syndromes (HIGM). This work is of interest to the readership of J Clin Medicine, including this reviewer, a hematologist/oncologist. Of special potential interest are the molecular mechanisms described in the manuscript.
However, the quality of the manuscript could be improved by consideration of the following issues.
Major issues:
in the experimental section 2.3, lines 107-111, authors stated that PCR products of VH3-23 were generated using Pfu polymerase followed by TA cloning/sequencing. Of note, Pfu polymerase generates blunt-end PCR products that are not expected to function well with TA cloning. The authors need to clarify this important technical aspect. In the Introduction, it would be helpful if a formal classification of HIGM syndromes were included. An example given on the website of the National Organization for Rare Disorders (https://rarediseases.org/rare-diseases/hyper-igm-syndrome/ where HIGM syndromes are listed in numerical order from Type 1 through Type 5, with the X-linked hyper IgM syndrome listed separately. The authors should update their review of the literature to include an excellent recent review (Yazdani et al., Clin Immunology 198 (2019) 19-30). Included in the update should also be a discussion of the prevalence and epidemiology of HIGM syndromes for the general readership.
Minor issues:
issues with language that should be addressed, including syntax, the use of articles, plural versus singular, as well as punctuation. The use of abbreviations throughout the manuscript is inconsistent, such as pt vs. patient. The use of plural vs. singular is inconsistent as well (PBMC vs. PBMCs). The authors also should introduce the full name when an abbreviation is first introduced. For example, the term “LPD” was never explained. It would be helpful to include a glossary with a list of all abbreviations. the clarity of the Tables needs to be improved. Specifically, Tables should not be split, or at the very least, the split Table needs to have its own header on the new page. The font size of the + (plus) and – (minus) signs is too small, especially for the minus signs. In Table 1, the intervals between words in the column on the left are inconsistent and make the Table difficult to read. The authors should refrain from centering or using full justification; for this column, left justification may solve the inconsistency in spacing between words. lymphocyte counts reported in Table 2 have inconsistent units of measure. For lymphocyte counts in the thousands, periods should not be used.
Specific minor issues:
The numbering of the Figures is inconsistent. In Line 260, the authors presumably mean (Figure 5) [not (Figure 4)]. Likewise, in line 270, it appears that the authors meant to refer to Figure 6 [instead of Figure 5].
Reviewer 2 Report
The paper titled “Clinical, immunological and functional characterization of 6 patients with very high IgM levels” by Gallo et al. characterizes six patients that exhibited high IgM levels. The authors used techniques including next-generation sequencing, quantitative RT-PCR, and immunophenotyping to show that high IgM levels, regardless of their origin, are associated with recurrent infections, malignancies, and chronic lung changes. Thus, patients with high IgM levels should be monitored closely for development of aforementioned conditions. The findings are interesting and well-supported.
My major concern with the manuscript is that its readability is poor, and there a number of syntax errors that distract the reader. My suggestion to the authors is to rigorously improve both syntax as well as coherence of the text. Additional suggestion is that the authors provide a list of abbreviations in the beginning of the manuscript. This will aid readers in understanding the text with lesser effort.
